# The cryo-EM structure of the SNX–BAR Mvp1 tetramer

Dapeng Sun [1,2,6], Natalia V. Varlakhanova[1,6], Bryan A. Tornabene[1], Rajesh Ramachandran[3], Peijun Zhang[2,4,5✉] & Marijn G.J. Ford [1✉]

Sorting nexins (SNX) are a family of PX domain-containing proteins with pivotal roles in trafficking and signaling. SNX-BARs, which also have a curvature-generating Bin/Amphiphysin/Rvs (BAR) domain, have membrane-remodeling functions, particularly at the endosome. The minimal PX-BAR module is a dimer mediated by BAR-BAR interactions. Many SNX-BAR proteins, however, additionally have low-complexity N-terminal regions of unknown function. Here, we present the cryo-EM structure of the full-length SNX-BAR Mvp1, which is an autoinhibited tetramer. The tetramer is a dimer of dimers, wherein the membrane-interacting BAR surfaces are sequestered and the PX lipid-binding sites are occluded. The N-terminal low-complexity region of Mvp1 is essential for tetramerization. Mvp1 lacking its N-terminus is dimeric and exhibits enhanced membrane association. Membrane binding and remodeling by Mvp1 therefore requires unmasking of the PX and BAR domain lipid-interacting surfaces. This work reveals a tetrameric configuration of a SNX-BAR protein that provides critical insight into SNX-BAR function and regulation.

[1] Department of Cell Biology, University of Pittsburgh School of Medicine, Pittsburgh, PA 15261, USA. [2] Department of Structural Biology, University of Pittsburgh School of Medicine, Pittsburgh, PA 15260, USA. [3] Department of Physiology & Biophysics, Case Western Reserve University School of Medicine, Cleveland, OH 44106, USA. [4] Division of Structural Biology, Wellcome Centre for Human Genetics, University of Oxford, Oxford OX3 7BN, UK. [5] Electron Bio-Imaging Centre, Diamond Light Sources, Harwell Science and Innovation Campus, Didcot OX11 0DE, UK. [6]These authors contributed equally: Dapeng Sun, Natalia V. Varlakhanova. ✉email: peijun@strubi.ox.ac.uk; marijn@pitt.edu

Sorting nexins (SNX) are a large and varied family of phox-homology (PX) domain-containing proteins with functions in membrane trafficking and remodeling, signaling, and organelle movement[1,2]. SNX–BAR proteins form a subfamily that is characterized by the presence of a membrane-remodeling or curvature-sensing Bin/Amphiphysin/Rvs (BAR) domain[3] in addition to the PX domain which, typically, is a lipid-binding module[4]. Mammalian cells have at least 12 SNX–BAR proteins and yeast have 7[5,6]. SNX–BAR proteins are involved in several cellular processes that depend upon membrane remodeling, including protein and lipid trafficking to and from the endosome, endocytosis, and autophagy[5–7]. Defects in SNX–BAR function are associated with tumorigenesis, neurodegenerative diseases, and cardiovascular defects[8,9].

All SNX–BARs homo- or heterodimerize via extensive inter-actions between their BAR domains. Current models for SNX–BAR-mediated membrane remodeling propose that both the PX and BAR domains have to be engaged with the membrane to ensure specificity and efficient binding. While some SNX–BAR proteins consist only of their PX–BAR modules, others, including those involved in retromer-mediated retrograde trafficking and autophagic processes[5,10] have, in addition, a low-complexity N-terminal region, the function of which is unclear. In the case of the endocytic SNX–BAR SNX9, this low-complexity region is involved in allosteric regulation of SNX9 membrane binding and remodeling activites[11].

Mvp1 is a poorly characterized yeast SNX–BAR protein that shares conservation with the mammalian SNX–BAR SNX8. Mvp1 was initially identified as a genetic interaction partner for the fungal dynamin superfamily protein Vps1[12] and is required for retrograde trafficking from the endosome[13–15]. Here, we use cryoelectron microscopy with single-particle averaging to deter-mine the structure of the fungal SNX–BAR Mvp1. Full-length Mvp1 is a tetramer consisting of two Mvp1 dimers that self-interact in such a manner that the concave lipid-engaging sur-faces of the BAR dimers are buried. Tetramerization depends on the presence of the Mvp1 N-terminal region, which is also required for Mvp1 sorting function in vivo. Mvp1 lacking the N-terminal region retains membrane-remodeling activity and exhibits enhanced membrane binding in vivo and in vitro. Together, this work reveals a mechanism of regulation of Mvp1 function by self-assembly.

## Results

### Mvp1 functions in retrograde trafficking from the endosome.
In cells, Mvp1 localized to the endosomal compartment in a PI3P-dependent manner (Supplementary Fig. 1a)[14]. Loss of Mvp1 results in several trafficking defects in the cell[13]. Previous findings and data mining from whole-genome synthetic genetic array screens[16,17] provide strong evidence for Mvp1 function in retromer-dependent retrograde trafficking from the endosome to the TGN (Supplementary Table 1). Cells lacking Mvp1 exhibited defects in retrograde transport, as assessed by a change in the steady-state subcellular distribution of the CPY receptor Vps10 (Supplementary Fig. 1b, c)[14]. In budding yeast, Vps10 is recycled from the endosome to the TGN prior to fusion of the maturing endosomal compartment with the vacuole. In Δmvp1 cells, we observed a significant increase in vacuolar membrane localization of EGFP-tagged Vps10 expressed from a plasmid, indicative of defective endosomal recycling (Supplementary Fig. 1b). Con-sistently, Δmvp1 cells therefore secreted CPY whereas wild-type (W303A) cells did not (Supplementary Fig. 1d), as assessed by colony immunoblotting, using an anti-CPY antibody.

In vitro, purified Mvp1, like SNX8 and several other SNX–BARs[18], bound to liposomes containing PI3P (Supplementary Fig. 2a, b). Liposome-binding properties were the same using Mvp1 obtained from two different purification strategies (Supplementary Fig. 2a, b). Mvp1 also deformed liposomes into tubules with mean and median diameters of 49.8 (±13.4) nm and 48 nm (Fig. 1a and Supplementary Fig. 2h). Mvp1 is therefore involved in membrane remodeling and cargo sorting at the endosome.

### Full-length Mvp1 is a tetramer.
Full-length Mvp1 was tetra-meric at 150 mM NaCl, as determined by multiangle light scat-tering coupled with size-exclusion chromatography (SEC-MALS) (Fig. 1b). Mvp1 remained tetrameric under both low (50 mM) and high (250 mM) salt conditions, at pH 7.4 (Supplementary Fig. 2c and Fig. 1b). At 50 mM NaCl, there was a slight leading shoulder, perhaps indicating some formation of larger oligomers. At 250 mM NaCl, the observed molecular weight was slightly lower than the predicted theoretical weight for the tetramer. As the column elution profiles at both 150 and 250 mM NaCl were indistinguishable (Fig. 1b and Supplementary Fig. 2c), we inter-pret this as the tetramer being in fast exchange, compared with the column separation time, with a small population of dimers at 250 mM NaCl. This will tend to decrease the observed molecular weight. This result also indicates that the tetramerization is based, at least in part, on ionic interactions. Indeed, at 250 mM NaCl and pH 6.5, Mvp1 was fully tetrameric (Supplementary Fig. 2d). The method of purification of Mvp1 had no effect on its tetra-meric state (Supplementary Fig. 2e).

### Mvp1 tetramerization depends on its N-terminal region.
Existing structures of SNX–BAR proteins reported to date are invariably dimeric, with dimerization occurring via an extensive BAR–BAR interface that is conserved in all BAR domain-containing proteins[18–21]. However, the constructs used for the structural analyses consisted of a minimal PX–BAR module, removing any low-complexity sequence that some SNX–BAR proteins have at their N-termini. We therefore generated an Mvp1 construct similar to those used for the structural analyses, encompassing only the PX–BAR module and omitting the N-terminal low-complexity sequence (Mvp1 Δ2-78 and Mvp1 Δ2-100). Both Mvp1 Δ2-78 (Supplementary Fig. 2f) and Mvp1 Δ2-100 (Fig. 1c) were fully dimeric in solution, as assessed by SEC-MALS. To confirm this, we engineered an Mvp1 construct containing a PreScission protease site after residue 99. Prior to digestion, the protein behaved similarly to the wild type at 250 mM NaCl. After removal of the N-terminal sequence by digestion, the cleaved protein was dimeric (Supplementary Fig. 2g). Compared with full-length Mvp1, Mvp1 Δ2-100 dis-played increased lipid binding in vitro (Fig. 1d, e). Moreover, Mvp1 Δ2-100 retained an ability to tubulate liposomes in vitro (Fig. 1f), generating tubules with mean and median diameters of 72.9 (±9.9) nm and 73.5 nm, respectively (Supplementary Fig. 2h). Morphologically, tubules formed by Mvp1 Δ2-100 appeared to be more regular than those formed by Mvp1 but occasionally we observed liposomes that had been deformed into apparent helices (Fig. 1f inset). It is unclear why tubules generated by Mvp1 Δ2-100 are wider than those generated by Mvp1 (P < 0.0001, Mann–Whitney test) but it may reflect different modes, extents or regularity of assembly on the lipid surface.

When expressed in cells, Mvp1 Δ2-100 was localized to puncta (the endosomal compartment), similar to Mvp1, but also displayed an enhanced localization to the vacuolar membrane (Supplementary Fig. 3a, b). We interpret this result as an impairment in Mvp1 disassembly from endosomal membrane prior to fusion with the vacuole. However, Mvp1 Δ2-100 did not rescue the CPY trafficking defect in Δmvp1 cells as assessed by CPY secretion (Supplementary Fig. 3c). Taken together, these

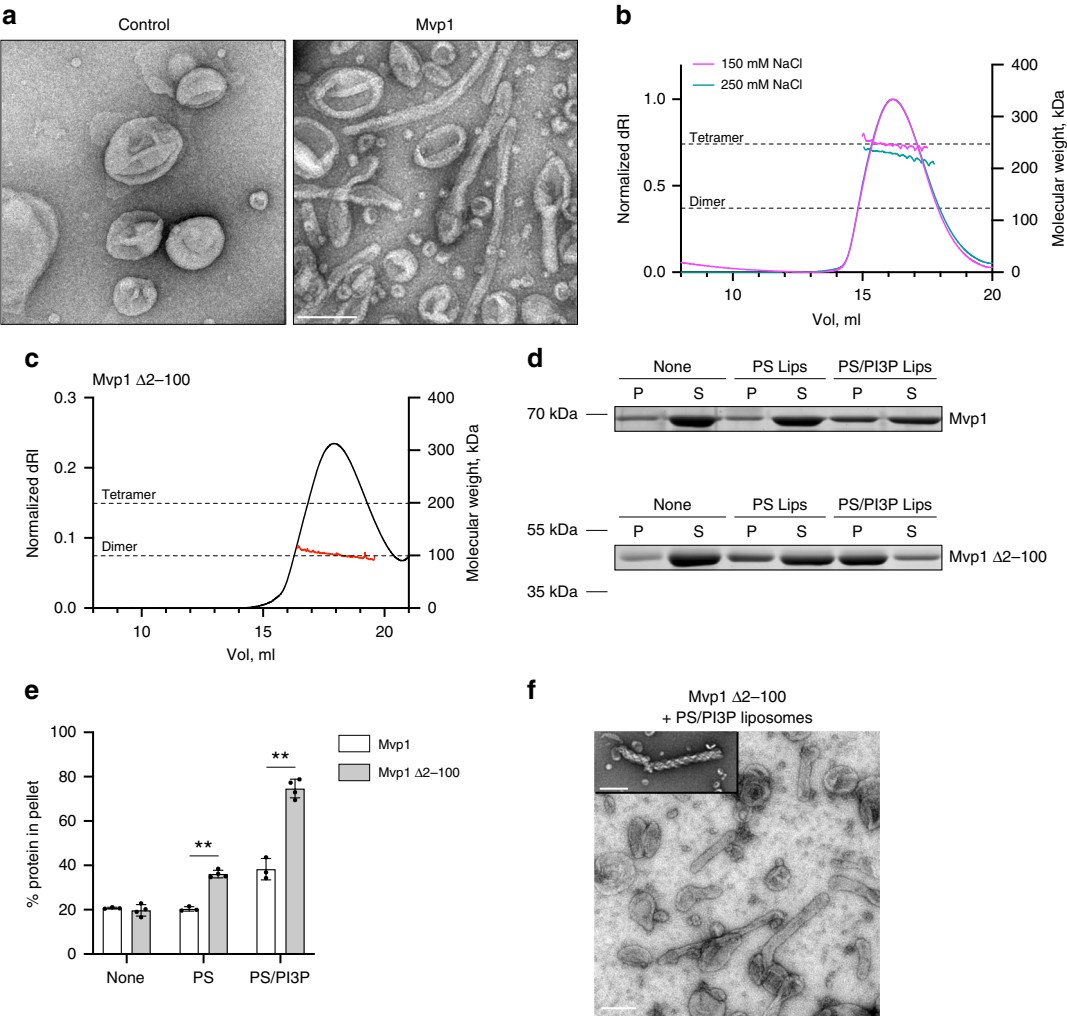

**Fig. 1 Mvp1 tetramerization is dependent on its N-terminus. a** Mvp1 deforms PS liposomes containing 5% PI3P into irregular tubes. Representative micrographs of negative-stained liposomes incubated either with buffer (control) or with Mvp1. Scale bar—200 nm. **b** Mvp1 is a tetramer. The absolute molecular weight of Mvp1 was determined using SEC-MALS. The molecular weight, shown in magenta or teal across the elution peaks, are plotted on the right-hand axis. Theoretical molecular weights of Mvp1 dimers and tetramers, calculated according to the Mvp1 sequence, are shown as dotted gray lines. The differential refractive index of the elutions are plotted on the left-hand axis. Mvp1 was eluted in buffer containing either 150 mM (magenta) or 250 mM (teal) NaCl. **c** As in **b**, Mvp1 Δ2-100 was eluted in buffer containing 250 mM NaCl. **d** Comparison of liposome binding by Mvp1 and Mvp1 Δ2-100. Protein (1.2 μM) was incubated without or with DOPS (PS) or DOPS + 5% PI3P (PS/PI3P) liposomes for 30 min at 21 °C prior to sedimentation. Shown is a representative result. P pellet, S supernatant. Positions of molecular weight markers are shown to the left of the gels. **e** Quantification of the results presented in **d**. Individual data points and mean ± s.d. are shown. $n = 3$ for Mvp1 and 4 for Mvp1 Δ2-100. A 3 × 2 factorial ANOVA was conducted to determine the effects of N-terminus (Mvp1 or Mvp1 Δ2-100) or the absence or presence of liposomes on the % sedimentation of protein. There was a significant interaction term ($P < 0.0001$) and each main effect was also significant ($P < 0.0001$). Selected pairs of values significant (Tukey HSD) at the 1% level are shown (**). For **d** and **e**, source data are provided as a Source Data file. **f** As in **a** but with Mvp1 Δ2-100. The inset shows an example of a rare subpopulation of highly twisted tubes observed only when PS/PI3P liposomes were incubated with Mvp1 Δ2-100. Scale bars—200 nm.

observations indicate that, while Mvp1 Δ2-100 retained membrane interaction and remodeling activity, it lost its function in endosomal trafficking and sorting in vivo.

**The cryo-EM structure of the Mvp1 tetramer.** We determined the structure of the Mvp1 tetramer using single-particle cryo-EM (Fig. 2a and Supplementary Figs. 4 and 5). Cryo-EM 2d classification of Mvp1 particles resulted in tetramer and some dimer classes (Supplementary Fig. 4a). The local resolution of the final map varied from ~4.2 Å (within the cores of the BAR domain dimers) to ~7.2 Å (the tips of the BAR domains) (Supplementary Fig. 5b). An atomic model for the Mvp1 tetramer (Fig. 2b) was generated using the SNX–BAR structures of *Homo sapiens* SNX9

(PDB 3DYT)[20] and *Chaetomium thermophilum* Vps5 (PDB 6H7W)[21] as a starting point (Supplementary Fig. 5c, d). These share 16 and 13% sequence identity with Mvp1 across the PX–BAR module. The tetramer consists of a tight embrace of the two SNX–BAR dimers, with the two concave faces of the BAR homodimers facing each other, slightly rotated with respect to one another, around a central axis through the middle of the BAR dimer, such that the distal tips of the homodimers pack adjacent to one another (Fig. 2c and Supplementary Fig. 6a). This packing serves to occlude and sequester the positively charged, concave, lipid-binding surface of each BAR dimer (Fig. 2d). In this configuration, the PX domain is unable to accommodate PI3P due to steric clashes with the *trans* BAR dimer in the tetramer (Supplementary Fig. 6b), as shown by superposition of the PI3P-bound

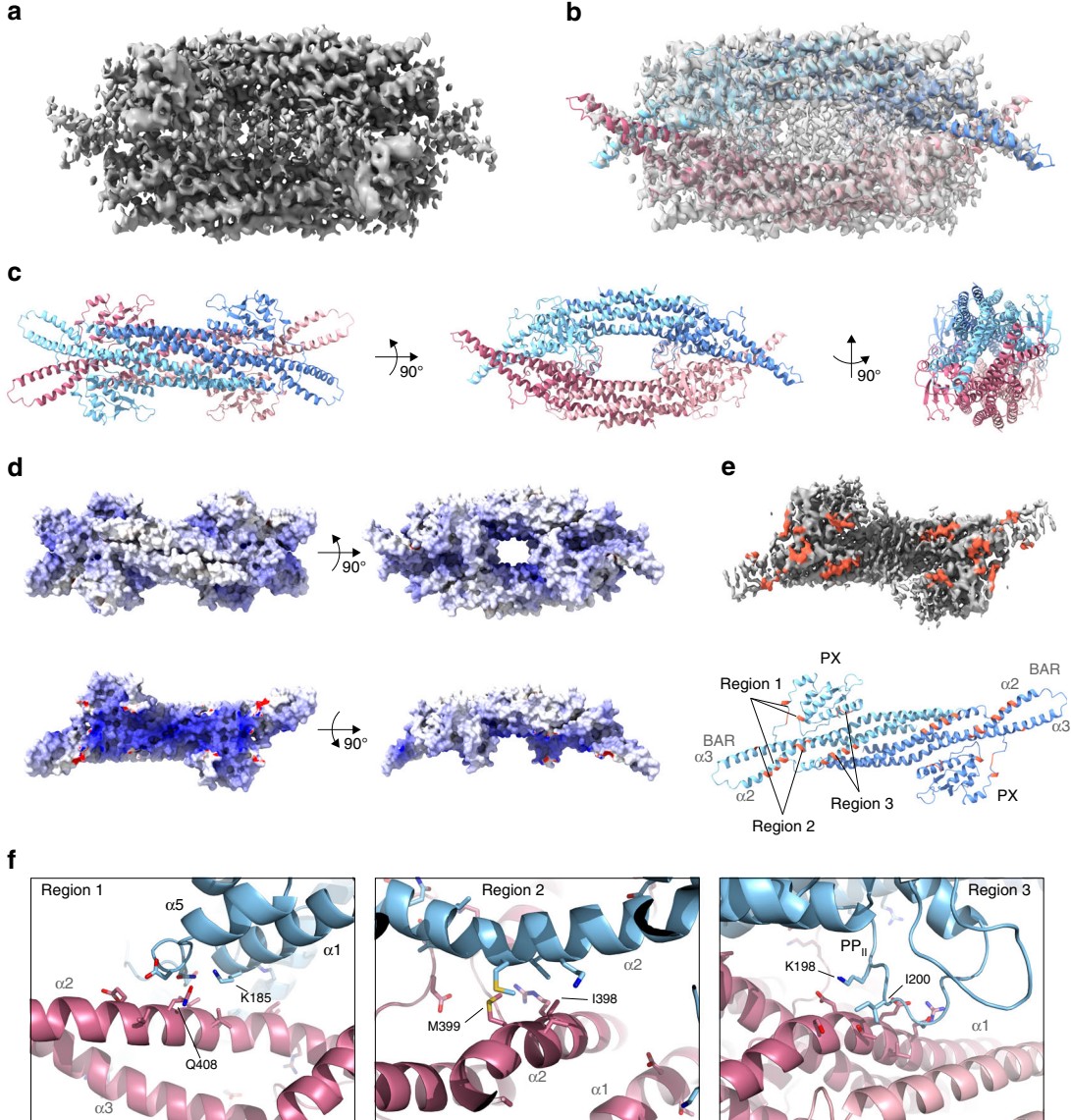

**Fig. 2 The cryo-EM structure of the full-length Mvp1 tetramer. a** Single-particle cryo-EM reconstruction of the Mvp1 tetramer. The D2-symmetrized, sharpened ($B = -252$ Å$^2$) map is shown at a contour level of 4σ. **b** Map, as in **a**, overlaid with a real-space refinement model of the Mvp1 tetramer. The upper Mvp1 SNX–BAR dimer is colored lighter and darker blue while the lower dimer is colored lighter and darker pink, to facilitate differentiation of the monomers. **c** Ribbon diagrams of the Mvp1 tetramer, colored as in **b**, viewed along each of the three twofold axes in the structure. **d** Surface models of the Mvp1 tetramer colored by electrostatic potential, colored from red (negative) to blue (positive). The scale is $-25$ to 25 $kT/e$. Upper row: the Mvp1 tetramer. Lower row: an isolated Mvp1 dimer. Note the sequestration of the positive charged concave surfaces into the interior of the tetramer. **e** Contact sites between the Mvp1 dimers responsible for tetramerization. Upper: map segment corresponding to the Mvp1 dimer with the contact sites between dimers colored in tomato. The map segment, extracted from the map presented in **a** and contoured at 4σ, is oriented with the concave surface of the BAR dimer facing the reader. Lower: ribbon model of the Mvp1 dimer with the contact sites colored in tomato, oriented as for the map segment. **f** Details of the three tetramerization regions. In these cases, the *trans* dimer is also shown, in pink, as in **c**.

PX domain from p40$^{phox}$[22] onto the Mvp1 SNX–BAR PX domain. Tetramerization occurs via three main regions. First, residues from the C-terminal end of helix α1 and the loop downstream of helix α5 in the PX domain (part of the Yoke in the SNX9 SNX–BAR structure) contact the distal end of BAR helix α2 from the *trans* Mvp1 dimer (Fig. 2e, f). Second, the C-termini and N-termini of BAR helices α2 and α3 interact around the twofold long axis of the tetramer (Fig. 2e, f). Third, there are close contacts between the top of the PX PP$_{II}$ loop and helix α1 from a BAR domain in the *trans* BAR dimer (Fig. 2e, f). Together, these regions generate a tetramerization interface of 1278 Å$^2$ at each end of the tetramer, forming a combined interface area of 2556 Å$^2$. For comparison, the constitutive BAR–BAR interface is 6083 Å$^2$.

**Mutations within the PX domain disrupt Mvp1 tetramerization.** We next sought to disrupt tetramerization by targeting an assembly interface. We therefore generated Mvp1 K198A, Mvp1 R199A, I200A, and Mvp1 $^{198}$KRI→AAA (hereafter Mut1) (Supplementary Fig. 6c). These residues lie within the PP$_{II}$ loop and form part of Region 3 of the tetramerization interface (Fig. 2f). In addition, K198 is predicted to line the back of the PI3P-binding pocket[20,22]. By SEC-MALS, Mut1 is entirely dimeric (Fig. 3a). Mvp1 K198A and Mvp1 R199A, I200A exhibited an apparent intermediate molecular weight (Fig. 3b). As before, we interpret this as being due to a mixed population of rapidly interconverting dimers and tetramers, perhaps as these intermediate mutations partially destabilize the tetramer, rather

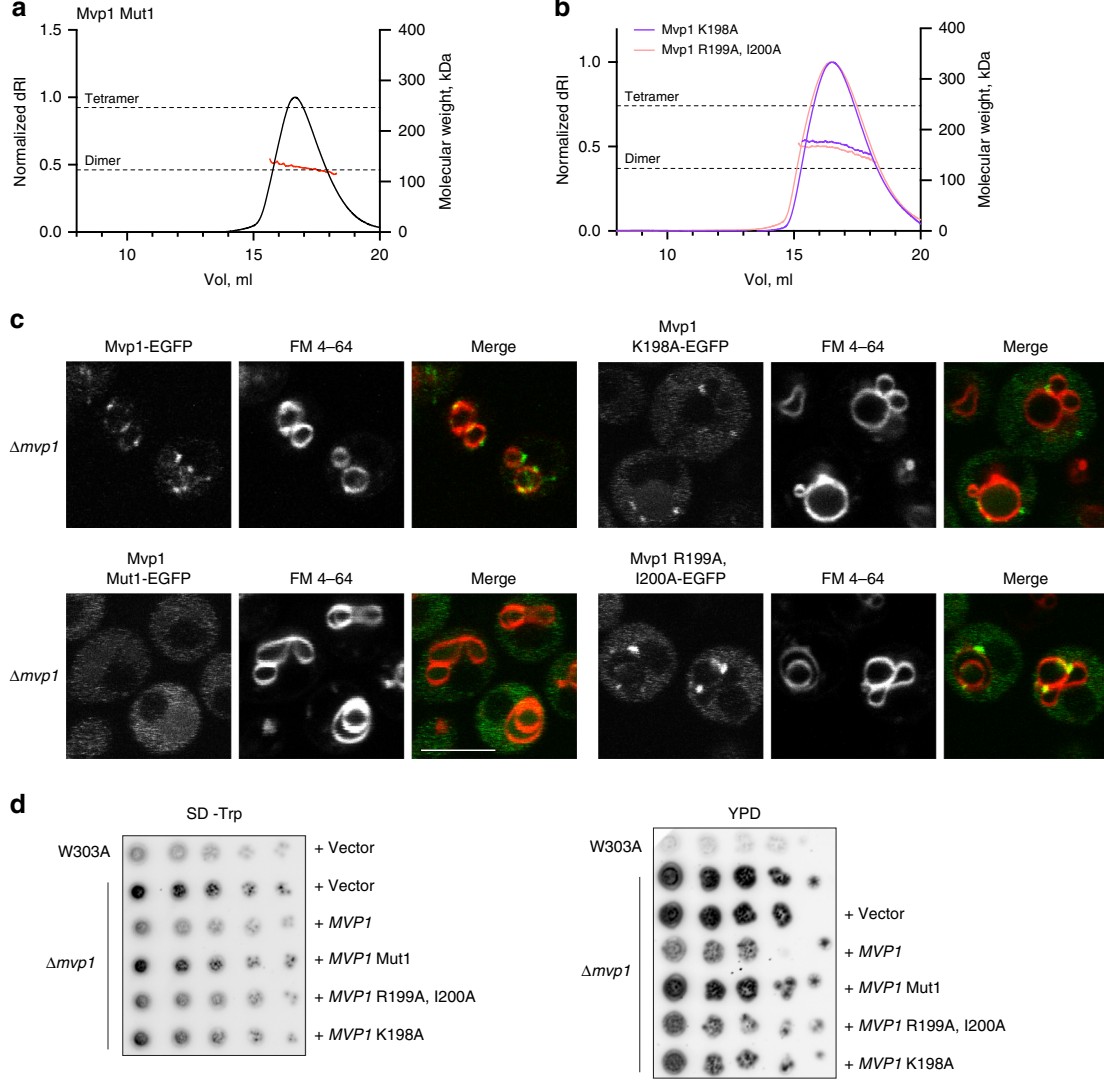

**Fig. 3 Mvp1 Mut1 is dimeric and is defective in lipid binding and CPY sorting in vivo.** The absolute molecular weights of Mvp1 Mut1 (**a**) or Mvp1 K198A (purple) or Mvp1 R199A, I200A (pink) (**b**) were determined using SEC-MALS as in Fig. 1. Proteins were eluted in buffer containing 250 mM NaCl. **c** Subcellular localization of EGFP-tagged Mvp1 and its mutants in Δ*mvp1* cells. Vacuoles were stained with the lipophilic dye FM 4–64. Images were obtained by confocal microscopy. Scale bar—5 μm. **d** CPY secretion assays. W303A or Δ*mvp1* cells expressing vector or the indicated construct were plated onto SD-Trp to maintain plasmid selection (left) or YPD (right) plates and incubated at 30 °C for 24 h. Plates were then overlaid with nitrocellulose for an additional 16 h. CPY secretion was detected by immunoblotting using an anti-CPY antibody. The leftmost spot in each case is 2 μl of a OD$_{600}$ = 0.5 culture: spots to the right of this are sequential fivefold dilutions. For **d** source data are provided as a Source Data file.

than fully, as in the case of Mut1. When expressed in cells, while Mvp1 K198A and Mvp1 R199A, I200A maintained some membrane association (punctate), Mvp1 Mut1 was cytosolic (Fig. 3c). Consistently, Mvp1 K198A and Mvp1 R199A, I200A partially retained CPY sorting function. Mvp1 Mut1, on the other hand, was completely defective, and as defective as cells lacking Mvp1 altogether (Fig. 3d).

**Single-particle cryo-EM analysis of Mvp1 Mut1.** Mvp1 Mut1 did not interact with, or tubulate, lipid templates as assessed using sedimentation assays and negative-stain electron microscopy (Fig. 4a–c). To verify that Mvp1 Mut1 retained its expected dimeric architecture, we used 2d and 3d classification of single particles by cryo-EM to analyze its structure (Fig. 4d, Supplementary Fig. 6d). It is clear from both the 2d class averages and the 3d alignments, although of intermediate resolution (~7.7–8.2 Å), that Mvp1 Mut1 is dimeric. Furthermore, the relative orientation of the PX domains with respect to its neighboring BAR

domains are also unaltered. We therefore conclude that Mut1 does not impair function by gross structural changes but rather by specifically disrupting membrane binding. As Mvp1 Mut1 is defective in both lipid binding and tetramerization, the question arises as to whether tetramerization is a prerequisite for lipid binding. Mvp1 Δ2-100 is dimeric and readily binds and remodels lipid templates. Hence, Mvp1 Mut1 has a compound defect in tetramerization and, independently, in lipid binding. The structure of the tetramer suggests that the concave BAR lipid-binding surface is sequestered and the PX lipid-binding pocket is inaccessible. Hence, we propose that tetramerization is likely to inhibit, rather than being a prerequisite for lipid binding.

## Discussion
Here, we present the structure for a tetrameric configuration of the SNX–BAR Mvp1, in which the lipid-binding interfaces are occluded by dimerization of SNX–BAR dimers. The low-complexity Mvp1 N-terminus plays an essential role in

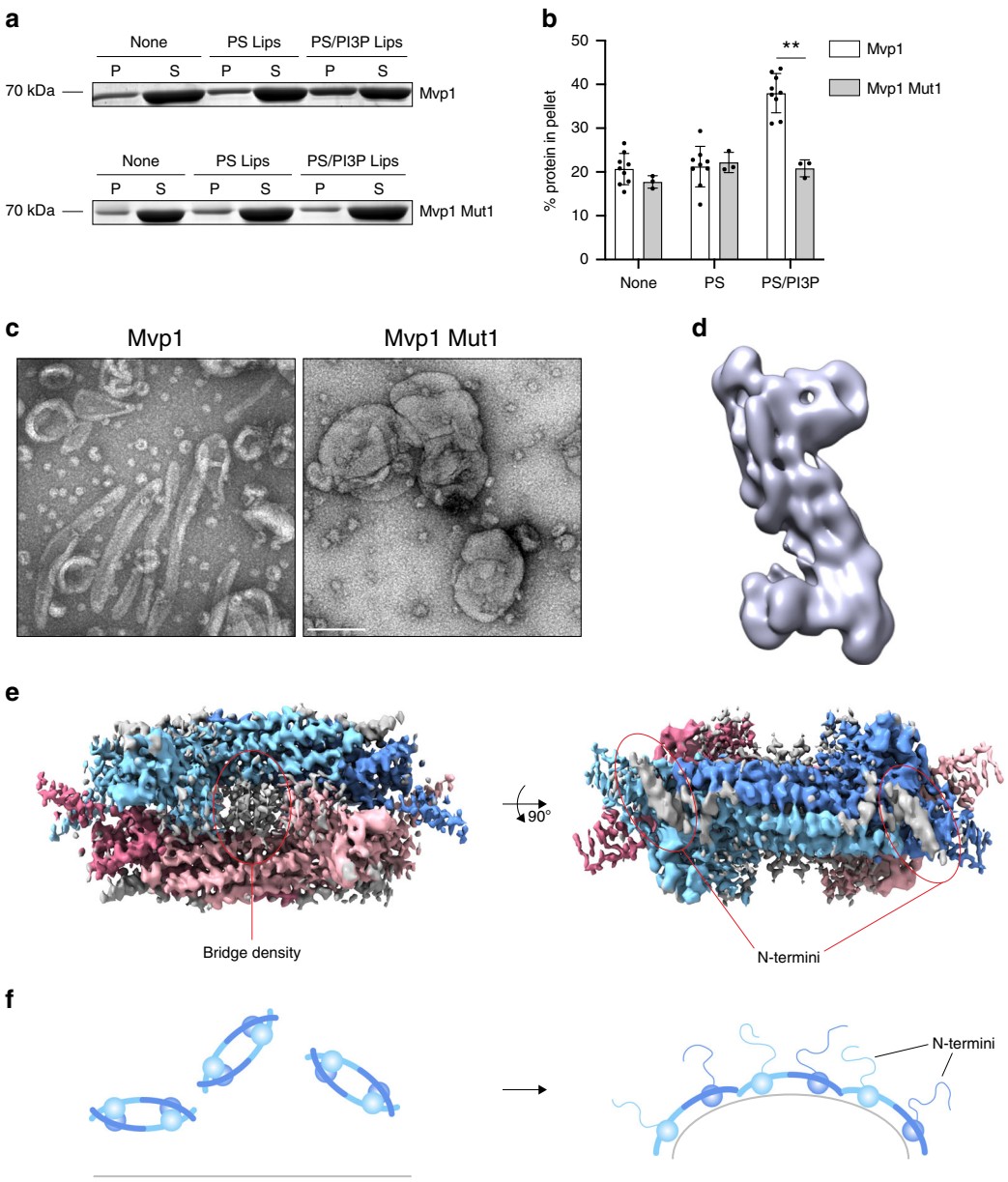

**Fig. 4 Mvp1 Mut1 is defective in membrane binding and remodeling in vitro. a** Comparison of liposome binding by Mvp1 and Mvp1 Mut1. Protein (1.2 μM) was incubated without or with DOPS (PS) or DOPS + 5% PI3P (PS/PI3P) liposomes for 30 min at 21 °C prior to sedimentation. A representative result is shown. P pellet, S supernatant. Positions of molecular weight markers are shown to the left of the gels. **b** Quantification of the results presented in **a**. Individual data points and mean ± s.d. are shown. $n = 9$ for Mvp1 and 3 for Mvp1 Mut1. A 3 × 2 factorial ANOVA was conducted to determine the effects of the mutation (Mvp1 or Mvp1 Mut1) or the absence or presence of liposomes on the % sedimentation of protein. There was a significant interaction term ($P = 0.0001$) and both effects of liposome and Mvp1 or Mvp1 Mut1 were also significant ($P < 0.0001$ and $P = 0.0002$). A selected pair of values significant (Tukey HSD) at the 1% level is shown (**). Please note that the same data set for Mvp1 was used to compare with Mvp1 Mut1 here as was used to compare the different Mvp1 purification protocols presented in Supplementary Fig. 2b. For **a** and **b**, source data are provided as a Source Data file. **c** Mvp1 Mut1 is unable to tubulate PS liposomes containing 5% PI3P. Representative micrographs of negative-stained liposomes incubated either with Mvp1 or with Mvp1 Mut1 are shown. Scale bar—200 nm. **d** Mvp1 Mut1 is dimeric. Representative cryo-EM 3d class average of Mvp1 Mut1, with an approximate resolution of 8 Å. **e** Unassigned density. The D2-symmetrized, sharpened map, at contour level 4σ, is colored according to distance from the atomic model of the Mvp1 tetramer: color is assigned if the density is within 3 Å of the atomic model. The unassigned density is therefore shown in gray. **f** Model of membrane remodeling by Mvp1.

tetramerization. While the N-terminus is unstructured and therefore not visible in our model, we do observe some unassigned density between the PX domains, crossing the divide between the opposed BAR dimers. We speculate that this density is contributed by the N-terminus. We term this bridge density (Fig. 4e). This assignment is consistent with a recent detailed characterization of SNX9[11]. SNX9 has an N-terminus that

consists of an SH3 domain followed by a low-complexity linker of ~193 residues. The linker harbors a short acidic stretch that directly interacts with the SNX9 BAR dimer[11], at a site that is equivalent to the region on the Mvp1 BAR domains that abut the bridge density, as assessed by hydrogen–deuterium exchange (Supplementary Fig. 6e). In addition, the Mvp1 N-terminus has a pI of 3.68 (residues 1–100) and contains several highly acidic

patches. Mvp1 lacking residues 1–100 has a calculated pI of 8.51 and the region where the bridge density docks is very positively charged. The bridge density may contribute to stabilization or formation of the tetramer. The Mut1 mutations that compromise tetramerization lie within the PX PP$_{II}$ loop. We note that the top of the PP$_{II}$ loop and the bridge density lie in close proximity. An attractive possibility is that the PP$_{II}$ loop plays a role in positioning of the N-terminus in such a way that tetramerization is promoted. In support of this, we note that several of the Mvp1 Mut1 2d classes appear to have a free, or unleashed, density adjacent to the PX–BAR dimer which may be contributed by the N-terminus. In any case, any additional stabilization from the N-terminus would enhance the total tetramerization interface. The bridge density may be contributed by an ordered binding motif within the Mvp1 N-terminus. The Mvp1 N-terminus is also essential for in vivo function. While Mvp1 lacking the N-terminus can remodel membrane in vitro, it cannot rescue CPY receptor sorting. It is therefore likely that the N-terminus binds to a factor essential for sorting, which may assist in opening of the tetramer, revealing the lipid-binding interfaces for membrane engagement and subsequent remodeling events (Fig. 4f).

Mvp1 shares homology with several mammalian SNX–BAR proteins. The highest conservation is with SNX8 and SNX7 (at ~22% identity across the PX–BAR module). SNX8 function is unclear but is involved in endosomal sorting[23]. Its inactivation is linked to the congenital cardiac defect Tetralogy of Fallot[24]. SNX7 has an unknown protein-sorting function but has been associated with Aβ processing[25]. Mvp1 also shares homology with SNX1 (17% identity). SNX1 heterodimerizes with either SNX5 or SNX6 to generate one of the membrane remodeling and cargo selecting modules that function with mammalian retromer[26], which has been linked to neurodegeneration, diabetes, pathogen invasion, and various cancers[1,8,27]. Unlike the conservation with SNX7 and SNX8, which is restricted to the PX and BAR domains, Mvp1 is conserved with SNX1 throughout its sequence, including the low-complexity N-terminus. Interestingly, it has been reported that a portion of the SNX1 pool in the cytosol of HeLa cells elutes from a size-exclusion column at a volume consistent with tetramers[28,29]. In addition, the F-BAR protein PACSIN/syndapin has been reported to form tetrameric barrel-like structures, as assessed by size-exclusion chromatography, cross-linking and negative-stain electron microscopy[30]. These observations suggest that tetramerization may be a conserved mechanism of SNX–BAR and some other BAR protein regulation.

Based on our data that the Mvp1 N-terminus is functionally important, we speculate that it will be a target for regulation, either by interaction with other proteins or by posttranslational modifications. Indeed, the NetPhos 3.1 server predicts several phosphorylation sites within the N-terminus[31] and it was reported that a lysine within the N-terminus is ubiquitylated[32]. In the case of SNX1 and SNX2, both of which have similar low-complexity N-termini, multiple phosphorylation sites have been identified in different phosphoproteome analyses[33–37]. In the absence of the Mvp1 N-terminus, we observe increased lipid binding suggesting that the N-terminus may have an autoinhibitory function. Autoinhibition in BAR domain-containing proteins has been reported[3]. For example, SNX9 is autoinhibited in solution and this has been shown to be due to the linker preceding the PX–BAR module[11]. Other examples include the PACSIN/syndapin F-BAR, the I-BAR IRSp53, and the Drosophila F-BAR protein Nervous Wreck[38–40]. In all these cases, inhibition is mediated by auxiliary regions or domains adjacent to the BAR. Here, we propose the N-terminus promotes tetramerization as a mechanism of autoinhibition.

Our sedimentation assays show that full-length Mvp1 interacts with liposomes, but only in the presence of PI3P. The structure of the tetramer suggests that the PX domains could not accommodate PI3P without steric clashes of the acyl chains with the trans BAR dimers, at least with the PX domains in the same positions as observed in the structure. Incubation of Mvp1 tetramers with the PI3P headgroup or DiC8 PI3P did not affect the tetramer, as assessed by SEC-MALS (data not shown). A membrane containing PI3P might therefore be required for productive disruption of the Mvp1 tetramer.

How could tetrameric Mvp1 interact with membrane containing PI3P at all? The structure we describe was obtained in the absence of lipid or lipid headgroups. In the presence of membranes, we observe PI3P-dependent lipid binding so the conformation of the PX domain relative to the trans BAR dimer has to change to accommodate the PI3P. This may lead to further opening of the tetramer. It has recently been reported that the SNX3 PX domain inserts its β1–β2 loop, as well as its membrane-inserting loop, into the bilayer when bound to membranes containing PI3P[41]. Mvp1 has a stretch of hydrophobic residues in its long β1–β2 loop. The equivalent of the membrane-inserting loop in Mvp1 is the top of the PP$_{II}$ loop, adjacent to Mut1, which we observe is critical for membrane binding. We therefore speculate that the initiating event in changing the conformation of the PX domain relative to the trans BAR dimer is insertion of the β1–β2 loop into the bilayer. This may be followed by PI3P binding and disruption of the tetramer. This would necessitate the presence of PI3P in a membrane context. The N-terminus may be displaced by sequential binding of the PX domain to PI3P followed by BAR interaction with the membrane's negative charge. In the cell, tetramer opening may be further enhanced by cargo binding or posttranslational modifications of the N-terminus.

SNX–BAR tetramerization adds a layer of complexity to the regulation of SNX–BAR function in membrane remodeling and in cargo sorting. As SNX–BAR proteins are vital in well-characterized machineries such as retromer, and as deficiencies in SNX–BARs are associated with a range of health challenges, these findings provide a precedent and foundation for future studies on the functional regulation of SNX–BAR proteins via self-assembly.

## Methods

**Yeast media**. YPD (1% yeast extract, 2% peptone, 2% glucose, and supplemented with tryptophan and adenine) was used for routine growth. Synthetic complete (SC; yeast nitrogen base, ammonium sulfate, 2% glucose, and amino acids) or SD (synthetic defined; as SC but lacking an appropriate amino acid) were used for growth prior to microscopy or to maintain plasmid selection. Diploids were sporulated by overnight incubation in YPA (1% yeast extract, 2% peptone, and 2% potassium acetate) followed by 1–2 days in SPO (1% potassium acetate, 0.1% yeast extract, and 0.05% glucose).

**Yeast genetic manipulation and molecular biology**. Strains of *Saccharomyces cerevisiae* used in this work are listed in Supplementary Table 2. ORF deletions were generated in W303A/α diploids by homologous recombination. Appropriate cassettes, flanked with sequence (30 nucleotides) proximal to the coding sequence of the target ORF, were amplified from pFA6a-His3MX6, to allow selection of diploids containing a modification by growth on SD-His. Diploids were then sporulated. Tetrads were manually dissected, and candidate knockout haploids were extensively validated.

**Cloning**. Plasmids used in this work are listed in Supplementary Table 3. *S. cerevisiae MVP1*, with flanking upstream and downstream regions of 250 and 150 nucleotides, was amplified from genomic DNA prepared from W303A/α diploids. All constructs were generated by splicing by overlap extension, if required, and Gibson assembly. Primers used in this work are listed in Supplementary Table 4

**Protein expression and purification**. Full-length Mvp1 (codons 1–511), its mutants and truncations were expressed from two vectors: pET-15b (Novagen), as N-terminal His$_6$ fusions followed by a thrombin cleavage site or from a homemade variant of pMW (a gift from Helen Kent), as a C-terminal MBP fusion preceded by a PreScission cleavage site. Constructs were expressed in *Escherichia coli* strain BL21-Codon Plus(DE3)-RIPL (Agilent). Cells were grown in 2xTY at 37 °C until mid-log phase, after which expression was induced by addition of IPTG (42.5 μM).

Proteins were expressed overnight at 21 °C. Cells were harvested and resuspended in TN150 buffer (20 mM TRIS/Cl pH 7.5, 150 mM NaCl, and 1.93 mM β-mercaptoethanol) before lysis by homogenization (Avestin C3; Emulsiflex). Lysates were then clarified (142,000 g at 4 °C for 45 min).

His$_6$-Mvp1 fusions were applied, in batch, to Ni-IDA resin (Macherey-Nagel). The resins were washed with TN150, loaded into a column and fusion proteins were eluted using a 0–250 mM imidazole gradient in TN150. Peak fractions were pooled and dialyzed into TN150. The dialysates were then applied to an anion exchange column. Proteins were eluted with a 150–500 mM NaCl gradient. Peak fractions were pooled, concentrated, and passed over a Superdex 200 size-exclusion column, equilibrated with TN250 (TN with 250 mM NaCl). Peak fractions were concentrated and flash frozen in liquid nitrogen for storage until use.

C-terminal MBP-tagged fusions were applied, in batch, to amylose resin (NEB). After washing in TN150, fusion proteins were eluted with 10 mM maltose in TN150. Peak fractions were pooled and dialyzed overnight into fresh TN150 at 4 °C, while being cleaved with PreScission protease (GE Healthcare). PreScission protease and cleaved MBP were removed by sequential incubations with glutathione sepharose (Macherey-Nagel) and amylose resin. Mvp1 proteins were then further purified by anion exchange and size-exclusion chromatography, as for the His6 fusions. A slight modification was used to purify Mvp1 Δ2-100. The uncleaved protein was purified in TN150 supplemented with 1 mM EDTA (TN150E) and 0.1 mM PMSF by amylose resin, elution, dialysis into TN150E, anion exchange, and size exclusion chromatography. Then the protein was cleaved in TN250 with PreScission, after which the PreScission was removed and the protein was passed over a Superdex 200 in TN250, before concentration and snap freezing.

**Preparation of yeast for microscopy.** Yeast were grown at 30 °C overnight in YPD or appropriate SD to maintain plasmid selection. Yeast were then diluted in YPD and grown to mid-log phase. Vacuolar membranes were stained with FM 4–64 (10 μM, Thermo Fisher Scientific) for 45 min, followed by washing and incubation in SC medium without dye for 1 h. Prior to imaging, cells were plated onto No. 1.5 glass-bottomed coverdishes (MatTek Corporation) previously treated with 15 μl 2 mg/ml concanavalin-A (Sigma-Aldrich).

**Imaging and image analysis.** A Nikon (Melville) A1 confocal, equipped with a 100× Plan Apo 100× oil objective, was used to obtain confocal images and Z-stacks. NIS Elements Imaging software was used to control acquisition. Images were processed using Fiji.

**CPY secretion assay.** After overnight growth to saturation in appropriate media at 30 °C, yeast were diluted and regrown to mid-logarithmic phase in YPD. Yeast were then diluted to 0.5 OD$_{600}$/ml and fivefold serial dilutions were made in water. Two microliters of each dilution was spotted onto YPD or SD-Trp plates after which they were incubated at 30 °C for 24 h. The colonies were overlaid with nitrocellulose, followed by additional incubation at 30 °C for 16 h. The membranes were then extensively washed with TBST (TRIS-buffered saline, supplemented with 0.1% Tween 20) and blocked for 1 h with TBST containing 5% bovine serum albumin. The membrane was probed for secreted CPY using an anti-CPY mouse monoclonal antibody (ab113685; Abcam) for 2 h at room temperature. The secondary antibody was IRDye 680RD-conjugated goat anti-mouse antibody (926-68070; LI-COR). The signal was detected using a ChemiDoc MP Imaging System (Bio-Rad).

**Preparation of liposomes.** Liposomes were made from 100% PS (1,2-dioleoyl-sn-glycero-3-phospho-L-serine; Avanti) or 95% PS + 5% PI3P (1,2-dioleoyl-sn-glycero-3-phospho-(1′-myo-inositol-3′-phosphate; Avanti)). Lipid mixtures were dried using a stream of nitrogen gas and were desiccated for at least 3 h. The lipids were rehydrated in 20 mM TRIS/Cl pH 7.5, 250 mM NaCl to a concentration of 1 mg/ml. Liposomes were extruded through a 100 nm Nucleopore membrane (Whatman).

**Liposome sedimentation assays.** Mvp1, its mutants or truncations were incubated in the absence or presence of 0.1 mg/ml 100 nm PS or PS/PI3P liposomes at 21 °C for 30 min in a final volume of 100 μl in TN250 and at a concentration of 1.2 μM. The samples were then centrifuged at 40,000 g for 30 min at 4 °C in the S55-A2 rotor (Thermo Fisher Scientific). Pellets and supernatants were separated and equal proportions of each were analyzed by SDS–PAGE. Bands were quantified by integration using Fiji.

**SEC-MALS.** Mvp1, its mutants or truncations were subjected to size-exclusion chromatography using a Superdex 200 10/300 gl column (GE Healthcare) equilibrated 20 mM TRIS/Cl pH 7.4, 250 mM NaCl, 1.93 mM β-mercaptoethanol, unless explicitly stated. In general, 500 μl of 20–21.2 μM protein was injected, with the exception of Mvp1 Δ2-78, where sample limitations permitted injection of 500 μl of 14 μM protein. The Mvp1 with an engineered PreScission protease cleavage site after residue 99 was passed onto the column uncleaved or after cleavage with 10 U of PreScission protease in TN buffer for 90 min before removal of the PreScission

protease by incubation with glutathione-sepharose beads. The column was coupled to a static 18-angle light-scattering detector (DAWN HELEOS-II) and a refractive index detector (Optilab T-rEX) (Wyatt Technology). Data were collected continuously at a flow rate of 0.3 ml/min. Data analysis was performed using the program Astra VII. Monomeric BSA (2.0 mg/ml) (Sigma) was used for data quality control.

**Negative-stain electron microscopy.** For tubulation assays, 10 μg liposomes were incubated with 10 μM (final concentration) Mvp1 or Mvp1 Mut1 (20 μl final volume) for 30 min at room temperature. 3 μl sample aliquots (protein with liposomes or liposomes alone) were adsorbed to glow-discharged 300-mesh carbon-coated copper grids and stained with 2% uranyl formate. Images were recorded on a Tecnai T12 Spirit electron microscope, operating at 120 kV with a LaB$_6$ electron source, at the indicated magnification on a 4,000 × 4,000 Gatan Ultrascan charge-coupled device camera.

**Cryo-EM sample preparation and imaging.** Purified Mvp1 or Mvp1 Mut1 (~5 μM protein) was applied to glow-discharged holey carbon grids (Quantifoil Cu R2/1, 300 mesh) and plunge-frozen in liquid ethane using an FEI Vitrobot Mark IV. Images were acquired at the University of Virginia School of Medicine Molecular Electron Microscopy Core on a Falcon 3EC detector in counting mode using an FEI Titan Krios at 300 kV with a nominal magnification of 75,000, corresponding to a final pixel size of 1.056 Å. For each image stack, a total dose of about 58 electrons per square angstrom was equally fractioned into 49 frames (~1.2 e$^-$/Å$^2$/frame). SerialEM was used for automated data collection[42]. Defocus values used to collect the data set ranged from −0.5 to −3.5 μm. Further details are given in Supplementary Table 5.

**Cryo-EM data processing.** For cryo-EM data, beam-induced motion correction was performed using MotionCor2[43] to generate dose-weighted averaged micrographs and dose-weighting micrographs from all frames. Contrast transfer function parameters were estimated using CTFFIND4[44] from averaged micrographs. Other procedures of cryo-EM data processing were performed within RELION 3.0[45] using the dose-weighted micrographs.

Approximately 6500 particles of Mvp1 were manually picked from ~100 lowpass-filtered micrographs using e2boxer.py from the EMAN2 suite[46] and were subjected to reference-free 2d classification in RELION. The best representative 2d class averages were selected as templates for automatic particle picking of 300 micrographs in RELION. After one round of 2d classification, ~20,000 particles were selected. The RELION implementation of the Stochastic Gradient Descent algorithm was used to generate a de novo 3d initial model from the selected particles. The generated low-resolution initial model and the selected 20,000 particles were subsequently used for further 3d classification and 3d auto-refinement, and finally converged to a map with resolution ~7 Å, determined using the gold-standard Fourier Shell Correlation (FSC) 0.143 criterion.

The 7 Å map was used as reference for further particle auto-picking from the whole data set. Approximately 1,230,000 particles were auto-picked from 1620 micrographs for further processing. The whole set of particles was cleaned to remove contaminants or junk particles by three rounds of 2d classification. Finally, ~200,000 particles were selected for further 3d reconstruction.

We observed an apparent D2 symmetry existing in the 3d reconstruction from the side view and the 2d classification results. In order to reduce the impact of imposition of symmetry on the reconstruction, therefore, we performed the 3d reconstruction procedures without and with different symmetries (Supplementary Fig. 4b).

For C1 symmetry, after 3d classification into five classes, the most populated class, which accounted for 39.8% of the data set (~82,000 particles), was used for 3d-masked auto-refinement. This yielded a map of ~6.1 Å resolution (gold-standard FSC 0.143 criterion). A 5.3 Å map was generated after sharpening with a B-factor of −190 Å$^2$ (post processing in RELION).

C2 symmetry was imposed on three user-defined axes (Fig. S4). For each axis, after 3d classification, the most populated class was selected for 3d-masked auto-refinement. This yielded three maps with the following resolutions:

4.6 Å (97,000 particles, after sharpening with a B-factor of −210.7 Å$^2$),
4.4 Å (91,000 particles, after sharpening with a B-factor of −220 Å$^2$),
4.7 Å (115,000 particles, after sharpening with a B-factor of −234 Å$^2$).

For the reconstruction with D2-imposed symmetry, after 3d classification (two rounds), a final map of 4.2 Å resolution was obtained after sharpening with a B-factor of −152.8 Å$^2$. This was used for subsequent model building and refinement.

Maps were visualized using ChimeraX[47] or Coot[48] and local map resolutions were calculated using RELION.

The same procedure was used for Mvp1 Mut1 data collection and processing. Due to the distribution of particle orientations (Supplementary Fig. 6d), we were unable to obtain high-resolution structures. We therefore subjected the particles to 2d and 3d classification analysis.

Data collection and processing statistics are summarized in Supplementary Tables 5 and 6.

**Model building**. The model for the Mvp1 tetramer was built with Coot[48], using an auto-sharpened map generated in Phenix[49]. The structures of *Homo sapiens* SNX9 (PDB 3DYT) and *Chaetomium thermophilum* Vps5 (PDB 6HZW) were used as a starting point (Fig. S5c, d). The map quality within the core of the BAR domains (Supplementary Fig. 5e) enabled verification of the correct sequence register. The model was completed using the density, secondary structure predictions generated using PSIPRED[50], local distance restraints generated with ProSMART[51], and existing SNX–BAR structures as guides. Some long loops could not be assigned well in the model including the β1–β2 loop in the PX domain, the loop connecting the PX domain to the BAR domain and the loop connecting BAR helices α2–α3. The model was refined against the 4.2 Å auto-sharpened map using phenix.real_space_refine. The final model statistics are listed in Supplementary Table 7.

The electrostatic potential of the surface of the Mvp1 tetramer was calculated, at a pH of 6.8, using APBS in the PDB2PQR server[52]. Sizes of tetramerization interfaces were calculated using the PISA web server[53].

**Reporting summary**. Further information on research design is available in the Nature Research Reporting Summary linked to this article.

## Data availability

Data supporting the findings of this paper are available from the corresponding authors upon reasonable request. A reporting summary for this article is available as a Supplementary Information file.

The cryo-EM map of Mvp1 is deposited in the Electron Microscopy Data Bank under accession code EMD-20555. The real-space-refined atomic model of the Mvp1 tetramer is deposited in the Protein Data Bank under accession code 6Q0X. The source data underlying Figs. 1d, e, 3d, 4a, b and Supplementary Figs, h, 3b, c are provided as a Source Data file.

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

## Acknowledgements

We thank Frances Alvarez for discussion, Kelly Dryden and the University of Virginia School of Medicine Molecular Electron Microscopy Core for assistance with data collection and Doug Bevan for computing support. Work in the Ford laboratory is supported by the National Institutes of Health grant GM120102 (M.G.J.F.). This work is also supported by the National Institutes of Health grant P50AI150481, the Regional Consortia for High-Resolution Cryoelectron Microscopy grant U24GM116790, the UK Wellcome Trust Investigator Award 206422/Z/17/Z, and the UK Biotechnology and Biological Sciences Research Council grant BB/S003339/1 (P.Z.). R.R. is supported by the National Institutes of Health grant GM121583. The Titan Krios and Falcon direct electron detector at the University of Virginia School of Medicine Molecular Electron Microscopy Core were obtained with NIH S10-RR025067 and S10-OD018149.

## Author contributions

N.V.V and M.G.J.F. conceived and designed the project; D.S., N.V.V., B.A.T., R.R., and M.G.J.F. performed the experiments; D.S., N.V.V., P.Z., and M.G.J.F. analyzed the data; N.V.V. and M.G.J.F. wrote the paper, with input from all authors.

## Competing interests

The authors declare no competing interests.
