## [Peer Review File · Nature Communications]

Reviewers' Comments:

Reviewer #1:

Remarks to the Author:

Sun, Varlakhanova and colleagues present a new structure of full-length yeast Mvp1 determined by cryoelectron microscopy (cryoEM) and single particle analysis (SPA). Mvp1 is a member of the sorting nexin (SNX) protein family possessing a canonical PI3P-binding phox-homology (PX) domain and a membrane-tubulating Bin/amphiphysin/rvs (BAR) domain. Other members of this family include the yeast Vps5 and Vps17 proteins associated with Retromer, and mammalian proteins such as SNX8 and SNX1 regulating endosomal trafficking. For context, my expertise is in structural biology of membrane trafficking proteins, but I am not a technical expert with cryoEM.

This is an interesting study that presents new insights into how SNX-BAR proteins can be functionally assembled. The data convincingly shows that Mvp1 is important for vacuolar/endosomal trafficking in yeast, and then uses biochemistry, biophysics and cryoEM to demonstrate that recombinant Mvp1 forms a stable tetramer in solution. The interesting aspect of this tetramer is that the core BAR-domain dimers are interlocked so that the expected membrane binding surfaces of the PX and BAR domains in Mvp1 are occluded. Also they show that the tetramer formation is controlled by the N-terminal disordered region, potentially via loose associations between dimers as judged by unmodeled cryoEM density, and that the N-terminus is required for the trafficking function of Mvp1. Overall I think the paper is well written, and have only some relatively minor questions and suggestions.

1. Is it possible to provide measurements of the tubule diameters formed by Mvp1 using the PC/PS/PI3P liposomes (Fig 1A) and also for the delta2-100 construct (Fig. 1F). Are there any differences in the tubule diameters?
2. At the end of the discussion it is stated that SNX1 is reported to form tetramers in cells. I think the best that can be said from the two references provided is that SNX1 runs on gel filtration with an apparent molecular weight that is consistent with tetramer formation, and the statement should reflect this uncertainty.
3. The Mvp1 tetramer structure shows that the putative PI3P-binding pocket in the PX domain would not be able to bind to the PI3P headgroup due to steric restraints. Have the authors measured the binding of the PI3P headgroup (e.g. by ITC) to the Mvp1 tetramer and Mvp1 delta2-100 dimer? If it binds to the dimer but not the tetramer, then this would confirm this hypothesis. In addition, could the addition of PI3P itself induce disruption of the tetramer into dimers?
4. Is anything known about post-translational modification (e.g. phosphorylation) of the Mvp1 N-terminus (or potentially of more highly studied related SNX-BAR proteins)? Is it possible that PTMs in this long disordered region could provide a cellular mechanism for tetramer-dimer exchange? I'm not suggesting extra experiments, but it might be worth discussing the possibility.

Brett Collins

Reviewer #2:

Remarks to the Author:

This study examines the mechanistic basis by which SNX-BAR proteins induce membrane tubulation. In particular, by studying the yeast SNX-BAR Mvp1, Sun and colleagues describe that the N-terminal low-complexity region of this protein is central to the formation of an auto-inhibited tetramer of the BAR domain dimer. Release of the tetramer leads to enhanced membrane association and increased tubulation activity. Overall, this is a really interesting study that combines cryo-EM structural analysis with in vitro reconstitution and in vivo phenotypic rescues to provide a very compelling argument for the proposed model. I very much recommend publication as the findings and concepts will be a wide interest to those researchers studying the evolutionary conserved function of SNX-BAR proteins in endosomal sorting.

Comments:

Given that the PI3P binding sites in the PX domains of the tetrameric form are occluded how do the authors consider that liposome binding is achieved in this autoinhibited state?

Does Mvp1 have a predicted amphipathic helix? If yes, then this should be highlighted and discussed within the structural context?

A key suggestion is that tetramerisation is likely to inhibit, rather than being a prerequisite for lipid binding. This conclusion is chiefly reached through comparison of the Mut1 and delta2-100 mutants. A cleaner mutagenic analysis would add to the support for this important conclusion. It is appreciated however, that this may be a difficult undertaking.

Is it possible to provide further validation that the assignment of the 'bridge density' to the N-terminal region is correct?

Finally, could the authors speculate on how the switch between tetramer and dimer is initiated?

Minor comments:

Mammalian retromer is not considered to form a functional pentameric complex like its yeast counterpart. Mammalian retromer is the VPS26:VPS35:VPS29 heterotrimer and this is functionally and biochemically distinct to the SNX1/SNX2/SNX5/SNX6 Membrane remodelling complex.

Consideration should be given to other studies that have suggested an autoinhibited state for BAR domain containing proteins. For example, Wang et al., 2009 studying paccin/syndapin, and Lo et al., on the analysis of SNX9.

The study of Halbach et al., 2007 on the BAR domain mediated tetramerisation of Paccin should also be considered.

Reviewer #3:

Remarks to the Author:

To the authors,

This work provides valuable structural insight into putative mechanism that regulates the organization and membrane remodeling activity BAR-PX protein Mvp1.

It reveals that deletion of a 100-residues stretch in the N-terminal (1-100) does not abolish membrane binding, rather improves it, suggesting it serves an auto-inhibitory function. Deletion of this motif also causes changes in the structures of tubulated liposomes, indicating that deletion changes how Mvp1 assembled on lipid membranes. They go on to show that Mvp1 retains its endosomal localization following deletion of 1-100, but also localizes to other intercellular sites, and that 1-100 is required for cellular function (CPY secretion). Cryo-EM analysis of the soluble form of Mvp1, which is uniquely tetrameric, reveals three tetramerization interfaces. Interestingly, unlike, 1-100 deletion, abolishing tetramer interface 3 impairs membrane binding and tubulation activity of Mvp1, and similar to the N-terminal 1-100, tetramerization is critical for CPY secretion.

In all, the structural information is convincing and supported by supplemental work. This is a concise and very well written article that reveals novel structural insight into a group of proteins that are rather overlooked. Although, despite clear and convincing evidence, there are some outstanding questions (some very minor) that should be addressed:

- What are the properties of FM 4-64 and why is it used?

- Line 573: there is a reference to 'red dotted circles' in S4, but these are absent in the figure.
- Line 418: Maps were visualized... "in/using" is missing.
- There is a considerable amount of protein in the pellet in the absence of liposomes? Is this due to Mvp1 aggregation/oligomerization?
- What is the rationale for using such small/highly curved LUVs? Does Mvp1 sense curvature?
- How was tubulation efficiency quantified?
- How is tetramerization related to PPII motif activity? Does PPII interactions regulate loop-bridge interactions and thus tetramerization?
- Is PPII activity involved in membrane binding?
- Lipid tubules formed by addition of 1-100 deletion mutant are referred to as 'helical', but the image is really low-res and small. Also the tubes look significantly more constricted than WT-decorated tubes (in the same image, Fig. 1). Was there any further investigation into this phenomenon, changes in outer diameter, protein organization, etc.?
- What is the reason for the discrepancy in Mut1 liposomes binding efficiency, comparing PS to PS/PI3P liposomes? Is there a phospholipid (PI3P) specific component to the role of 1-100?

Best,
Anna Sundborger-Lunna

Reviewer #4:

Remarks to the Author:

Professor Ford and colleagues are to be congratulated for making a new discovery about the structural properties of an SNX-BAR domain protein. Their characterization of the auto-inhibited tetramer is fundamentally important and will likely lead to an exciting new chapter in the study of regulated BAR and SNX-BAR domain activities because many members of this family of proteins also harbor low-complexity and intrinsically disordered motifs. I have only minor comments for the authors to consider.

1. The localization microscopy appears to me to be limited to the vacuolar compartment rather than the endosomal compartment.
2. There seems to be a mismatch between Fig. S1B and S1C. Both wild type and mutant seem to me to have VPS10-GFP foci on 100% of the vacuoles. I can't see how the mutant only has ~10% GFP positive vacuoles, given the image data in S1B.
3. I suggest using the same color code in Figures 2E and 2F for the respective monomers to make it easier for the reader to go back and forth between the global view in 2E and the zoom in for 2F.

Reviewers' comments:

Reviewer #1 (Remarks to the Author):

Sun, Varlakhanova and colleagues present a new structure of full-length yeast Mvp1 determined by cryoelectron microscopy (cryoEM) and single particle analysis (SPA). Mvp1 is a member of the sorting nexin (SNX) protein family possessing a canonical PI3P-binding phox-homology (PX) domain and a membrane-tubulating Bin/amphiphysin/rvs (BAR) domain. Other members of this family include the yeast Vps5 and Vps17 proteins associated with Retromer, and mammalian proteins such as SNX8 and SNX1 regulating endosomal trafficking. For context, my expertise is in structural biology of membrane trafficking proteins, but I am not a technical expert with cryoEM.

This is an interesting study that presents new insights into how SNX-BAR proteins can be functionally assembled. The data convincingly shows that Mvp1 is important for vacuolar/endosomal trafficking in yeast, and then uses biochemistry, biophysics and cryoEM to demonstrate that recombinant Mvp1 forms a stable tetramer in solution. The interesting aspect of this tetramer is that the core BAR-domain dimers are interlocked so that the expected membrane binding surfaces of the PX and BAR domains in Mvp1 are occluded. Also they show that the tetramer formation is controlled by the N-terminal disordered region, potentially via loose associations between dimers as judged by unmodeled cryoEM density, and that the N-terminus is required for the trafficking function of Mvp1. Overall I think the paper is well written, and have only some relatively minor questions and suggestions.

We thank Dr. Collins for his positive comments.

1. Is it possible to provide measurements of the tubule diameters formed by Mvp1 using the PC/PS/PI3P liposomes (Fig 1A) and also for the delta2-100 construct (Fig. 1F). Are there any differences in the tubule diameters?

We thank Dr. Collins for this suggestion. We have performed this analysis and now present the data as **Fig. S2H**, which we reproduce below. Full-length Mvp1 deformed liposomes into tubules with mean and median diameters of 49.8 (\pm 13.4) nm and 48 nm whereas Mvp1 Δ 2-100-generated tubules with mean and median diameters of 72.9 (\pm 9.9) nm and 73.5 nm respectively.

The difference perhaps reflects different modes or extents of assembly of the Mvp1 constructs on the lipid surface in the presence or absence of the N-terminal extension, which results in different curvatures.

2. At the end of the discussion it is stated that SNX1 is reported to form tetramers in cells. I think the best that can be said from the two references provided is that SNX1 runs on gel filtration with an apparent molecular weight that is consistent with tetramer formation, and the statement should reflect this uncertainty.

The reviewer is correct. We have modified the statement accordingly (~line 154).

3. The Mvp1 tetramer structure shows that the putative PI3P-binding pocket in the PX domain would not be able to bind to the PI3P headgroup due to steric restraints. Have the authors measured the binding of the PI3P headgroup

(e.g. by ITC) to the Mvp1 tetramer and Mvp1 delta2-100 dimer? If it binds to the dimer but not the tetramer, then this would confirm this hypothesis. In addition, could the addition of PI3P itself induce disruption of the tetramer into dimers?

We have attempted to address this point in several different ways.

First, we performed ITC on Mvp1 and Mvp1 Δ 2-100 with DiC8 PI3P as the ligand. The cell had 75 μ M protein and the ligand concentration in the syringe was 2 mM. While we did not detect binding of the lipid to full length Mvp1, we did detect a binding signal for Mvp1 Δ 2-100. However, the binding is weak and accurate determination of affinity was not possible: we chose our initial ligand and cell concentrations on the basis of the isolated PX domain from p40 as reported in Bravo *et al.*, 2001, where a 5 μ M affinity for DiC8 PI3P was observed.

Second, we performed microscale thermophoresis on Mvp1 and Mvp1 Δ 2-100 in the presence of DiC4 PI3P, where both proteins had been labeled and tagged with RED-N-hydroxysuccinimide. The data is reproduced below. It shows that Mvp1 Δ 2-100 interacts weakly with DiC4 PI3P with an affinity of \sim 270 μ M while the affinity of the wild type is at least an order of magnitude lower (\sim 3 mM).

We also attempted to disrupt the full-length Mvp1 tetramer by incubation with either IP3 (the headgroup) or with DiC8 PI3P. In both cases, SEC-MALS indicated no alteration in tetramerization.

The thermophoresis and SEC-MALS data show that binding of the headgroup alone is weak and not sufficient to disrupt the tetramer. Together, our data indicate that binding does need both a lipid surface and the specific headgroup, which would be consistent with the coincidence detection previously observed with SNX-BAR proteins (van Weering *et al.*, 2010).

Another possibility is that the affinity of the isolated Mvp1 PX domain for headgroup/lipid is greater than it is in the context of the Mvp1 SNX-BAR. Future work will assess this and whether and how an adjacent BAR domain may modify PX affinity for headgroup / lipid.

These data are preliminary so are not included in the manuscript.

4. Is anything known about post-translational modification (e.g. phosphorylation) of the Mvp1 N-terminus (or potentially of more highly studied related SNX-BAR proteins)? Is it possible that PTMs in this long disordered region could provide a cellular mechanism for tetramer-dimer exchange? I'm not suggesting extra experiments, but it might

be worth discussing the possibility.

The Uniprot entry for *Homo sapiens* SNX1 lists 4 phosphorylation sites within its N-terminal extension, identified in different phosphoproteome analyses (Dephoure *et al.*, 2008; Zhou *et al.*, 2012; Bian *et al.* 2014). Two additional sites are predicted on the basis of sequence conservation with SNX1 from *Mus musculus*, both of which were also identified by mass spectrometry (Huttlin *et al.*, 2010).

For SNX2, the situation appears to be similar, with 5 phosphorylation sites identified in its N-terminal extension, in either *Homo sapiens* or *Mus musculus* SNX2 (Villen *et al.*, 2007; Huttlin *et al.*, 2010; Dephoure *et al.*, 2008; Bian *et al.*, 2014; Zhou *et al.*, 2013).

The data for Mvp1 is less comprehensive at this time. However, K98, within the N-terminal extension and part of the region we have shown is important for tetramerization, is ubiquitinated (Swaney *et al.*, 2013). This is something we will follow up closely. It is of note that the NetPhos 3.1 server does predict several potential phosphorylation sites within the N-terminal extension, but these remain unverified experimentally and we have no data on this currently.

Regulation of tetramerization by modification of the N-terminal extension would be an attractive mechanism and we have addressed this in our discussion (~line 160).

Brett Collins

Reviewer #2 (Remarks to the Author):

This study examines the mechanistic basis by which SNX-BAR proteins induce membrane tubulation. In particular, by studying the yeast SNX-BAR Mvp1, Sun and colleagues describe that the N-terminal low-complexity region of this protein is central to the formation of an auto-inhibited tetramer of the BAR domain dimer. Release of the tetramer leads to enhanced membrane association and increased tubulation activity. Overall, this is a really interesting study that combines cryo-EM structural analysis with in vitro reconstitution and in vivo phenotypic rescues to provide a very compelling argument for the proposed model. I very much recommend publication as the findings and concepts will be a wide interest to those researchers studying the evolutionary conserved function of SNX-BAR proteins in endosomal sorting.

We thank the reviewer for the overall positive evaluation of our study.

Comments:

Given that the PI3P binding sites in the PX domains of the tetrameric form are occluded how do the authors considered that liposome binding is achieved in this autoinhibited state?

If the Mvp1 tetramer were completely inflexible, PI3P would indeed not be able to be accommodated in its PX domain binding pocket as its acyl chains would clash with the *trans* BAR domains in the tetramer. This is deduced from superposition of the PX domain of p40 bound to DiC4 PI3P (Bravo *et al.* 2001) on a PX domain in our tetramer. However, our sedimentation assays show that the full-length protein interacts with liposomes, but only in the presence of PI3P so PI3P clearly plays a role in release from the tetrameric state.

DiC8 PI3P alone, or its headgroup, is, however, not sufficient to disrupt the tetramer. We showed this by SEC-MALS using full length protein preincubated with the DiC8 PI3P or the headgroup. The ITC and thermophoresis we performed in response to reviewers' comments indicates that the affinity for DiC8 PI3P in isolation (ITC) or DiC4 PI3P (thermophoresis), not in the context of a membrane, is relatively low. These data are reported in response to Reviewer #1, above.

PI3P in the context of a membrane is therefore a key trigger to initiate binding. Our favored interpretation relies upon the β 1- β 2 loop in the Mvp1 PX domain. In the case of the PX domain of SNX3, this same loop was shown to insert into the membrane (Lenoire *et al.*, 2018). Mvp1 has a long β 1- β 2 loop (with an LLF patch) which may aid in weak membrane association. This may alter the conformation of the PX domain relative to the *trans* BAR domains, to permit PI3P binding in its pocket, after which full unravelling of the tetramer can begin.

We note that some of the unassigned density in our map lies above the β 1- β 2 loop in our model of the Mvp1 tetramer. We again speculate that this is part of the N-terminus and may limit the flexibility of the β 1- β 2 loop. It is of interest to note that Mvp1 Δ 2-100, which does not have an N-terminus, binds to PS liposomes, whereas full-length protein does not. Mvp1 Δ 2-100 is dimeric but increased flexibility of the β 1- β 2 loop may also play a role.

Mvp1 Mut1 has a broken PI3P binding pocket and thus is not able to interact with lipid at all (in the presence of the N-terminus).

We have extensively rewritten the discussion to include our speculation (from line 181).

Does Mvp1 have a predicted amphipathic helix? If yes, then this should be highlighted and discussed within the structural context?

We assessed the sequence of Mvp1 with two amphipathic helix prediction softwares: HeliQuest and AmphipaSeek (Gautier *et al.*, 2008, Sapay *et al.*, 2006).

HeliQuest was previously used (van Weering *et al.*, 2012) to identify a putative amphipathic helix in SNX8. The identified region (residues 180 to 197 of SNX8) is upstream of the BAR domain. Sequence homology with Mvp1 would place the equivalent sequence of Mvp1 within the PX domain. The identified region covers a small helix and flanking sequence in our structure downstream of PX helix α 3. By HeliQuest, this region in Mvp1 has a very low amphipathic moment (unlike the case for SNX8). AmphipaSeeK does not highlight this Mvp1 sequence as an amphipathic helix and the most hydrophobic residues in this region – Trp249 and Tyr255 – face the BAR of the *trans* dimer.

Interestingly, Mvp1 has a helix, identified by HeliQuest, with a respectable hydrophobic moment. This runs from 114-131 and includes the helix at the N-terminus of our structure that we observe (Fig. 4E). AmphipaSeeK did not assess this to be an amphipathic helix, however.

The SNX9 amphipathic helix is upstream of the start of the density observed in the crystal structure (Pylypenko *et al.*, 2007). Immediately after the site of the amphipathic helix, density is observed that lies adjacent to the linker connecting the PX and BAR domains such that sequence from the N-terminus and linker form a parallel β -sheet. We do not have sufficient density to accommodate this in our structure and the corresponding sequence, immediately upstream of our PX model, appears to be a helix (as 114-131) which, as mentioned above, does not appear to be amphipathic.

As we are uncertain whether Mvp1 has an amphipathic helix and, if it does, where it is, we prefer not to speculate on this without more clarity.

A key suggestion is that tetramerisation is likely to inhibit, rather than being a prerequisite for lipid binding. This conclusion is chiefly reached through comparison of the Mut1 and delta2-100 mutants. A cleaner mutagenic analysis would add to the support for this important conclusion. It is appreciated however, that this may be a difficult undertaking.

We agree with both points the reviewer makes: that it would be great and that it would be difficult!

In order to bolster our conclusions, we have performed several additional experiments.

First, we engineered a PreScission site into the Mvp1 sequence between the N-terminal extension and the PX-BAR module. We purified this protein and analyzed its multimerization by SEC-MALS before and after digestion with PreScission protease. The results are now presented in the manuscript as **Fig. S2G** and are reproduced below. Prior to digestion, the protein runs mostly as a tetramer (as before, we use 250 mM NaCl which results in a small component of dimers). After digestion, all the protein is dimeric. Therefore, we conclude that the N-terminus is required to form and to maintain the tetramer.

Second, in an attempt to identify cleaner mutations, we made a number of mutations within the BAR domain. Those we have tried to date (K308A Q312A H315A and H315A F319A) did not exhibit significant differences when compared to wild-type protein in lipid binding properties (as assessed by liposome sedimentation assays) or in tetramerization (as assessed by SEC-MALS), with the exception that the wild-type protein had a definite upper size whereas the mutants both tended to larger sizes at the front of the peak (please see image below). As design of these particular mutants was based on regions of close approach of the BAR domain and the “bridge density” (which we could not interpret, and which has been symmetrized), this involved a bit of guesswork and we may not have gotten it quite right. Our future work will focus on targeted disruption of more defined tetramerization interfaces and key residues involved in their formation (Regions 1-3), as well as the PX β 1- β 2 loop, which may be involved in contacting the bridge density.

Is it possible to provide further validation that the assignment of the ‘bridge density’ to the N-terminal region is correct?

Any unambiguous assignment will require additional structural work. One approach will be to make Fab fragments against the N-terminus. Another will rely upon co-crystallization attempts with the Mvp1 PX-BAR module and peptides from the N-terminus. This work is ongoing.

However, we have several lines of indirect evidence that suggest that this density does indeed belong to the Mvp1 N-terminus.

The first and strongest is a comparison to a recent detailed characterization of the “endocytic” SNX-BAR SNX9 (Lo *et al.*, 2017). SNX9 has an N-terminal SH3 domain connected to the PX-BAR module via a low-complexity linker of ~193 residues. The authors demonstrated that an “acidic stretch” within the linker can directly interact with PX-BAR module. Furthermore, hydrogen-deuterium exchange showed that most protected patch on the PX-BAR module lies within the region of the BAR dimer which is exactly where, in the case of Mvp1, the bridge density is observed. We

have discussed this in the reworked text (~line 126) have introduced a new figure (**Fig. S6E**, reproduced below) to convey this.

On several levels, the SNX9 linker and the Mvp1 N-terminal extension are similar. Lo *et al.* named the linker binding site (confirmed by mutagenesis) the “acidic stretch”. The Mvp1 N-terminus is extremely acidic. Residues 1-100 have a pI of 3.68. The protein as a whole has a pI of 5.35 and the Mvp1 Δ 1-100 has a pI of 8.53. The Mvp1 N-terminus has several very acidic stretches and we are of course currently analyzing those.

We also note that Mvp1 containing a PreScission site is dimeric after cleavage. We observe that some of the 2d classes of the Mvp1 Mut1 (which is all dimeric) have what appear to be “unleashed” N-termini (red arrows in **Fig. S6D**) and no density at an equivalent position as we observed within the tetramer.

We have done some additional characterization, albeit indirect. SEC-MALS of wt Mvp1 shows that it is tetrameric at 150 mM NaCl. Mvp1 Mut1 is entirely dimeric. By analytical size exclusion, we observe that increasing the salt in the buffer to 500 mM moves the elution peak of wt Mvp1 to elute at the same volume as Mvp1 Mut1 (hence dimeric).

We pooled the peaks eluting at high salt and re-dialyzed the protein into 150 mM NaCl before passing it over the column at 150 mM NaCl once again. This shifts the peak back to an elution volume earlier than the Mvp1 Mut1 dimer.

We therefore believe that the ability to tetramerize is an inherent ability of the protein and is defined by electrostatic interactions, consistent with the pI analysis of differing regions. The presence of a part of the N-terminus within the unassigned density would be consistent with this.

Finally, could the authors speculate on how the switch between tetramer and dimer is initiated?

This comment is closely related to the first comment about possible mechanisms of liposome binding. We addressed this point in detail together with the first comment in our response above.

Minor comments:

Mammalian retromer is not considered to form a functional pentameric complex like its yeast counterpart. Mammalian retromer is the VPS26:VPS35:VPS29 heterotrimer and this is functionally and biochemically distinct to the

SNX1/SNX2/SNX5/SNX6 Membrane remodelling complex.

Thank you. We have reworded the relevant segment of text (~line 150).

Consideration should be given to other studies that have suggested an autoinhibited state for BAR domain containing proteins. For example, Wang et al., 2009 studying pacsin/syndapin, and Lo et al., on the analysis of SNX9.

The study of Halbach et al., 2007 on the BAR domain mediated tetramerisation of Pacsin should also be considered.

Thank you. We have addressed these oversights and included these contributions in our discussion.

Reviewer #3 (Remarks to the Author):

To the authors,

This work provides valuable structural insight into putative mechanism that regulates the organization and membrane remodeling activity BAR-PX protein Mvp1.

It reveals that deletion of a 100-residues stretch in the N-terminal (1-100) does not abolish membrane binding, rather improves it, suggesting it serves an auto-inhibitory function. Deletion of this motif also causes changes in the structures of tubulated liposomes, indicating that deletion changes how Mvp1 assembled on lipid membranes. They go on to show that Mvp1 retains its endosomal localization following deletion of 1-100, but also localizes to other intercellular sites, and that 1-100 is required for cellular function (CPY secretion). Cryo-EM analysis of the soluble form of Mvp1, which is uniquely tetrameric, reveals three tetramerization interfaces. Interestingly, unlike, 1-100 deletion, abolishing tetramer interface 3 impairs membrane binding and tubulation activity of Mvp1, and similar to the N-terminal 1-100, tetramerization is critical for CPY secretion.

In all, the structural information is convincing and supported by supplemental work. This is a concise and very well written article that reveals novel structural insight into a group of proteins that are rather overlooked. Although, despite clear and convincing evidence, there are some outstanding questions (some very minor) that should be addressed:

We thank Dr. Sundborger-Lunna for comments, helpful suggestions and recommendations.

- What are the properties of FM 4-64 and why is it used?

FM 4-64 is a lipophilic dye that fluoresces when inserted into a lipid bilayer and is tool frequently used to label the vacuolar membrane in yeast *in vivo* (Vida and Emr, 1995). Labeling the vacuolar membrane with FM 4-64 is facile as the dye is endocytosed. With a chase, it reaches the vacuolar membrane and cleanly labels only that compartment as the end-point of the endocytic pathway.

- Line 573: there is a reference to 'red dotted circles' in S4, but these are absence in the figure.

Apologies. This has been amended and we have indicated the axes highlighted by the circles more clearly.

- Line 418: Maps were visualized... "in/using" is missing.

We have amended this. Thank you.

- There is a considerable amount of protein in the pellet in the absence of liposomes? Is this due to Mvp1 aggregation/oligomerization?

The reviewer is right: the protein that pellets in the absence of liposomes is consistently ~20% of the total amount of protein used in the assay and this is something we repeatedly observed. We invariably pass the protein through a 0.22 μm cellulose filter immediately prior to use in the sedimentation assays (as well as other assays) but this could still represent some aggregation/oligomerization.

To assess this, we repeated the sedimentation in the absence of liposomes and examined the protein at the bottom of the tube by negative stain electron microscopy. In general, the protein appeared to be monodisperse with little aggregation. We observe no oligomers or filaments. We include a representative micrograph below. In some regions, concentrated areas of stain also exhibited concentrated protein. This may reflect issues with staining or be indicative of a light burden of aggregation. We observe this same burden of sedimentation in the absence of liposomes regardless of which Mvp1 construct we use. Perhaps the consistent ~20% sedimentation we observe in the absence of lipid reflects non-specific binding to the tubes.

• What is the rationale for using such small/highly curved LUVs? Does Mvp1 sense curvature?

We compared liposomes that were extruded through 100 or 200 nm filters. We detected better interactions with this assay with liposomes extruded through a 100 nm filter. We therefore selected these to perform our experiments. This does indeed suggest that Mvp1 may sense curvature, but we have not yet rigorously assessed this. It has been proposed that SNX-BARs with amphipathic helices preferentially interact with smaller liposomes (those with a smaller radius of curvature) due to the greater ease of inserting an amphipathic helix into the outer leaflets of the bilayers of these vesicles. We discussed the potential of Mvp1 to harbor an amphipathic helix in response to reviewer #2. It is, however, unclear if Mvp1 has one.

• How was tubulation efficiency quantified?

We thank the reviewer for picking this up. Liposome binding was quantitated. Tubulation, where assessment is based on inspection of EM projection images, was qualitative, even though we remain of the impression that Mvp1 $\Delta 2-100$ is indeed more active in tubulation. We were hesitant to quantify this as there are too many potential caveats in the quantification and interpretation (such as perhaps tubulated liposomes preferentially adhering to the grid *etc.*).

We have rephrased the text to make this very clear (lines 52-53).

• How is tetramerization related to PPII motif activity? Does PPII interactions regulate loop-bridge interactions and thus tetramerization?

We suspect that this is indeed how the regulation will work. First, we find that Mut1, which consists of 3 mutated residues within the PPII loop, is entirely dimeric and the two intermediate mutations are in fast exchange between dimer and tetramer. Furthermore, the top of the PPII loop (as well as the $\beta 1-\beta 2$ loop in the PX domain, which is especially long in Mvp1) are in close proximity to the additional density.

We have added some discussion on this point and the following point in our revised discussion (~line 187).

• Is PPII activity involved in membrane binding?

The PII loop contains one of the key residues required to line the back of the PI3P pocket. Its mutation severely impairs membrane interaction. Therefore, the same residues that appear to be involved in tetramerization are also involved in membrane binding.

• Lipid tubules formed by addition of 1-100 deletion mutant are referred to as 'helical', but the image is really low-res and small. Also the tubes look significantly more constricted than WT-decorated tubes (in the same image, Fig. 1). Was there any further investigation into this phenomenon, changes in outer diameter, protein organization, etc.?

Both images shown in Fig. 1F in this case are of the structures we observe when we incubate PS/PI3P liposomes with Mvp1 Δ 2-100. Mvp1 wild type is shown in Fig. 1A. We included the small image as the inset to Figure 1F to mention that these twisted and helical structures are something we only occasionally see under our conditions of incubation of PS/PI3P liposomes with Mvp1 Δ 2-100: by far the majority of structures we observe are the tubes seen in the main part of Fig. 1F.

We are uncertain as to what those helical structures are, and we are of course interested in assessing whether we can tweak our conditions to enrich for those. It may be that these are formed when the Mvp1 Δ 2-100 assembles into extended oligomers on the lipid surface but this is maybe fanciful speculation!

We have reworked the legend to figure 1 to clarify this.

• What is the reason for the discrepancy in Mut1 liposomes binding efficiency, comparing PS to PS/PI3P liposomes? Is there a phospholipid (PI3P) specific component to the role of 1-100?

Mvp1 Mut1 does not interact with liposomes *in vitro*, with or without PI3P. The residues of Mut1 are critical for any lipid binding, regardless of whether the SNX-BAR is tetrameric or dimeric. In our hands, we observe no lipid binding at all with this mutant (or with K198A alone: Mut1 is K198A, R199A, I200A). The Mvp1 N-terminus is present in this construct.

Mvp1 Δ 2-100 exhibits much increased binding to PS / PI3P liposomes (WT: 39% sedimentation; Mvp1 Δ 2-100: 75% sedimentation). However, it also exhibits increased binding to PS liposomes (WT: no binding in excess of liposome-free control; Mvp1 Δ 2-100: 36% sedimentation). This construct of course lacks the N-terminus.

We therefore favor an explanation that depends on the Mvp1 N-terminus. In its absence, binding to PS liposomes becomes possible. We do not know whether this is via the PX domain (and its PI3P pocket) or via other interactions. Our currently untested hypothesis is that the Mvp1 β 1- β 2 loop (which is known to insert into the membrane when SNX3 binds PI3P (Lenoire *et al.*, 2018), can do so in the case of Mvp1 constructs that lack the N-terminus. If the Mvp1 has an N-terminus (such as Mvp1 Mut1), binding is absolutely dependent on an intact PI3P binding pocket.

We have introduced discussion on this point (~line 187).

Best,
Anna Sundborger-Lunna

Reviewer #4 (Remarks to the Author):

Professor Ford and colleagues are to be congratulated for making a new discovery about the structural properties of an SNX-BAR domain protein. Their characterization of the auto-inhibited tetramer is fundamentally important and will likely lead to an exciting new chapter in the study of regulated BAR and SNX-BAR domain activities because many members of this family of proteins also harbor low-complexity and intrinsically disordered motifs. I have only minor comments for the authors to consider.

We thank the reviewer for the positive assessment of our work.

1. The localization microscopy appears to me to be limited to the vacuolar compartment rather than the endosomal compartment.

Mvp1 distribution to the endosomal compartment, which presents as puncta in close proximity to the vacuolar membrane, has been previously observed and reported (Chi *et al.*, 2014). We also confirmed that these puncta are endosomal by colocalization with the FYVE domain of EEA1, a commonly used marker for the endosomal compartment in yeast. We include these data below. We did not include these data in our manuscript because this was not a novel observation.

2. There seems to be a mismatch between Fig. S1B and S1C. Both wild type and mutant seem to me to have VPS10-GFP foci on 100% of the vacuoles. I can't see how the mutant only has ~10% GFP positive vacuoles, given the image data in S1B.

We have rewritten this part to clarify what we were assessing in this case. In both cases (WT and $\Delta mvp1$ cells), Vps10-EGFP localizes to the perivacuolar endosomal compartment, as puncta, as previously reported (Chi *et al.*, 2014). The difference we were quantifying in S1B and S1C was in the colocalization of Mvp1 with the vacuolar membrane (labeled with FM 4-64). Vps10-EGFP only labels the vacuolar membrane if it is not efficiently sorted from the endosome prior to fusion of the endosome with the vacuole. Vps10-EGFP colocalization with FM 4-64 therefore is a readout for defects in endosomal sorting of Vps10. In our work, Vps10-EGFP localization to the vacuolar membrane is much more pronounced in the case of cells lacking Mvp1 and many more vacuoles exhibit this staining than in WT cells, but, in both cases, punctate endosomal staining of Vps10-EGFP persists.

We apologize for the confusion and have tried to improve clarity here.

3. I suggest using the same color code in Figures 2E and 2F for the respective monomers to make it easier for the reader to go back and forth between the global view in 2E and the zoom in for 2F.

Fig. 2E illustrates the regions on the Mvp1 SNX-BAR dimer where interactions with the other dimer occurs. We only show one dimer, which is shown in lighter and darker blues, as throughout this work. In Fig. 2F, we show the details of those binding sites in the context of the tetramer, where the *trans* dimer is depicted in lighter and darker pinks, as throughout this work. We have reworked the legend to clarify this.

Reviewers' Comments:

Reviewer #1:

Remarks to the Author:

I now recommend publication. The authors have performed some new and detailed experiments, as well as rewritten certain passages that address all of my previous requests. I have read the comments and the response for other reviewers as well, and believe that the authors have adequately addressed all major concerns.

Brett Collins

Reviewer #2:

Remarks to the Author:

The authors have adequately address my raised comments. Congratulations on an informative and important study.

Peter Cullen

Reviewer #3:

Remarks to the Author:

This is a well-written and concise report of an elegant set of experiments that reveal novel and critical mechanisms underlying the membrane remodeling activity of BAR-PX protein Mvp1. This work provides valuable structural insight into molecular mechanisms that regulate the organization and membrane remodeling activity of Mvp1. It reveals that deletion of a 100-residues stretch in the N-terminal (1-100) does not abolish membrane binding, rather improves it, suggesting it serves an auto-inhibitory function, which is supported by previous work by Rao et al, 2010. Deletion of this motif also causes changes in the structures of tubulates liposomes, indicating that deletion changes how Mvp1 assembled on lipid membranes. They go on to show that Mvp1 retains its endosomal localization following deletion of 1-100, but also localizes to other intercellular sites, and that 1-100 is required for cellular function (CPY secretion). Cryo-EM analysis of the soluble form of Mvp1, which is uniquely tetrameric, reveals three tetramerization interfaces. Interestingly, unlike, 1-100 deletion, abolishing tetramer interface 3 impairs membrane binding and tubulation activity of Mvp1, and similar to the N-terminal 1-100, tetramerization is critical for CPY secretion. In all, the structural information is convincing and supported by supplemental work. This is a concise and very well written article that reveals novel structural insight into a group of proteins that are rather overlooked. I enthusiastically support the publication of this study in Nature Communications and trust it will be highly interesting to a broad cell- and structural biology community, as well as biophysicists.

We thank Drs. Collins, Cullen, Sundborger-Lunna and the anonymous reviewer for the time and care taken in reviewing our manuscript, and for the helpful and insightful suggestions, which have resulted in a considerably strengthened manuscript.

Marijn G. J. Ford
01/29/2020

REVIEWERS' COMMENTS:

Reviewer #1 (Remarks to the Author):

I now recommend publication. The authors have performed some new and detailed experiments, as well as rewritten certain passages that address all of my previous requests. I have read the comments and the response for other reviewers as well, and believe that the authors have adequately addressed all major concerns.

Brett Collins

Reviewer #2 (Remarks to the Author):

The authors have adequately address my raised comments. Congratulations on an informative and important study.

Peter Cullen

Reviewer #3 (Remarks to the Author):

This is a well-written and concise report of an elegant set of experiments that reveal novel and critical mechanisms underlying the membrane remodeling activity of BAR-PX protein Mvp1. This work provides valuable structural insight into molecular mechanisms that regulate the organization and membrane remodeling activity of Mvp1. It reveals that deletion of a 100-residues stretch in the N-terminal (1-100) does not abolish membrane binding, rather improves it, suggesting it serves an auto-inhibitory function, which is supported by previous work by Rao et al, 2010. Deletion of this motif also causes changes in the structures of tubulates liposomes, indicating that deletion changes how Mvp1 assembled on lipid membranes. They go on to show that Mvp1 retains its endosomal localization following deletion of 1-100, but also localizes to other intercellular sites, and that 1-100 is required for cellular function (CPY secretion). Cryo-EM analysis of the soluble form of Mvp1, which is uniquely tetrameric, reveals three tetramerization interfaces. Interestingly, unlike, 1-100 deletion, abolishing tetramer interface 3 impairs membrane binding and tubulation activity of Mvp1, and similar to the N-terminal 1-100, tetramerization is critical for CPY secretion. In all, the structural information is convincing and supported by supplemental work. This is a concise and very well written article that reveals novel structural insight into a group of proteins that are rather overlooked. I enthusiastically support the publication of this study in Nature Communications and trust it will be highly interesting to a broad cell- and structural biology community, as well as biophysicists.